# Quorum Sensing in *Halorubrum saccharovorum* Facilitates Cross-Domain Signaling between Archaea and Bacteria

**DOI:** 10.3390/microorganisms11051271

**Published:** 2023-05-12

**Authors:** Thomas P. Thompson, Alessandro Busetti, Brendan F. Gilmore

**Affiliations:** Biofilm Research Group, School of Pharmacy, Queen’s University Belfast, 97 Lisburn Road, Belfast BT9 7BL, UK

**Keywords:** quorum sensing, archaea, biofilm, N-acyl homoserine lactones, halophiles

## Abstract

Quorum Sensing (QS) is a well-studied intercellular communication mechanism in bacteria, regulating collective behaviors such as biofilm formation, virulence, and antibiotic resistance. However, cell–cell signaling in haloarchaea remains largely unexplored. The coexistence of bacteria and archaea in various environments, coupled with the known cell–cell signaling mechanisms in both prokaryotic and eukaryotic microorganisms and the presence of cell–cell signaling mechanisms in both prokaryotic and eukaryotic microorganisms, suggests a possibility for haloarchaea to possess analogous cell–cell signaling or QS systems. Recently, N-acylhomoserine lactone (AHL)-like compounds were identified in haloarchaea; yet, their precise role—for example, persister cell formation—remains ambiguous. This study investigated the capacity of crude supernatant extract from the haloarchaeon *Halorubrum saccharovorum* CSM52 to stimulate bacterial AHL-dependent QS phenotypes using bioreporter strains. Our findings reveal that these crude extracts induced several AHL-dependent bioreporters and modulated pyocyanin and pyoverdine production in *Pseudomonas aeruginosa*. Importantly, our study suggests cross-domain communication between archaea and bacterial pathogens, providing evidence for archaea potentially influencing bacterial virulence. Using Thin Layer Chromatography overlay assays, lactonolysis, and colorimetric quantification, the bioactive compound was inferred to be a chemically modified AHL-like compound or a diketopiperazine-like molecule, potentially involved in biofilm formation in *H. saccharovorum* CSM52. This study offers new insights into putative QS mechanisms in haloarchaea and their potential role in interspecies communication and coordination, thereby enriching our understanding of microbial interactions in diverse environments.

## 1. Introduction

Quorum Sensing (QS) is a cell-to-cell communication process that allows bacteria and other microorganisms to share information and collectively modify their behaviors in response to changes in their environment, cell density, and species composition. This is achieved through the production, detection, and response to extracellular signaling, which leads to coordinated gene expression across the microbial population [1,2]. Initially, three QS systems were identified: Autoinducer-1 (AI-1), which is found primarily in Gram-negative bacteria but also in cyanobacteria [3] and archaea [4]; the oligopeptide-two-component-type QS, which is employed by Gram-positive bacteria; and Autoinducer-2 (AI-2), which relies on a furanosyl borate diester as a universal signal for communication between Gram-positive and Gram-negative bacteria. However, other QS systems in bacterial species have been described, including Autoinducer-3 (AI-3) [5] and 2-heptyl-3-hydroxy-4(1H)-quinolone in *Pseudomonas aeruginosa* (PQS) [6,7].

This LuxI/LuxR paradigm is widely conserved across Gram-negative bacteria. Because of this, proteins found in other AI-1-based systems have been termed LuxI-type synthases and LuxR-type receptors. The majority of AIs involved in these systems are N-acyl homoserine lactones (AHLs). AHL molecules are produced through a reaction between S-adenosyl methionine and an acylated acyl carrier protein (acyl-ACP), which is catalyzed by the LuxI homolog synthase. The acyl-ACP can vary, leading to the production of different AHL signaling molecules with acyl chains ranging from 4 to 18 carbon atoms [8]. LuxR-type receptors are selective for different AHLs, allowing cells to activate the transcription of specific genes in a controlled manner in response to environmental changes. Examples of phenotypic characteristics that can be controlled through QS include bioluminescence in *Vibrio harveyi* and *V. fischeri* [9], swarming motility in *Serratia marcescens* [10], biofilm formation in *Acinetobacter baumannii* [11], and virulence production in *P. aeruginosa* [12]. Over the last several years, research on QS systems has significantly advanced, uncovering new signaling molecules and mechanisms in various microorganisms; this includes not only bacteria but also fungi [13]. The increasing body of knowledge has shed light on the complexity and diversity of QS systems across different domains of life.

Given that bacteria and archaea co-inhabit various environments, such as marine environments, wastewater treatment plants, the mammalian digestive tract, and microbial mats [14], it is plausible that haloarchaea may possess QS systems that share common features with known bacterial systems. Recent studies in QS systems in archaea have identified the presence of the AI-2 system in the hyperthermophilic archaeon *Pyrococcus furiosus* [15], suggesting that archaea may use similar signaling molecules to communicate. Additionally, the methanogenic archaeon *Methanosaeta harundinacea* was found to produce carboxylated AHL molecules that function as QS signals and modulate gene expression [4]. Comparative genomics studies have identified the presence of LuxR-type transcriptional regulators in the Euryarchaeota, TACK, and DPANN lineages [16], hinting at possible QS systems in these organisms [14]. These LuxR proteins are predicted to bind various AHLs and non-AHL signaling molecules, suggesting potential interactions between haloarchaea and bacteria through QS-mediated communication. However, the specific roles and mechanisms of QS in archaea and our understanding of QS systems in lesser-studied groups, such as haloarchaea, remain limited [17].

To tackle this knowledge gap, we aim to investigate the potential presence and role of QS systems in haloarchaea, with a specific focus on *Halorubrum saccharovorum* CSM52. The hypothesis of this study is that *H. saccharovorum* CSM52 produces extracellular signaling molecules that can activate bacterial AHL-dependent QS phenotypes, indicating a possible involvement in interspecies communication and coordination. Furthermore, we will attempt to link this extract activity with a phenotype associated with AI production and investigate the potential ecological implications and functional roles of these signaling molecules in the context of haloarchaeal biology. By expanding our understanding of QS systems in haloarchaea, we hope to uncover novel communication mechanisms and their potential implications for complex microbiomes, thereby informing further studies in biotechnology and environmental sciences.

## 2. Materials and Methods

### 2.1. Bioreporter Strains and AHL Standards

The six QS-reporter strains used in this study were as follows: *Chromobacterium subtsugae* CV026, which was cultured in LB medium at 28 °C and maintained using 25 μg/mL kanamycin; *Escherichia coli* JM109 pSB536, cultured in LB medium at 30 °C and maintained with 25 μg/mL ampicillin; *E. coli* JM109 pSB401, cultured in LB medium at 37 °C and maintained with 20 μg/mL tetracycline; *E. coli* JM109 pSB1142, cultured in LB medium at 37 °C and maintained with 20 μg/mL tetracycline (all strains were kindly provided by Paul William from the School of Life Sciences at the University of Nottingham). Additionally, *Agrobacterium tumefaciens* NTL4(pZLR4) ATCC BAA-2240 was cultured at 28 °C in LB medium and maintained with 25 μg/mL gentamicin, while *Pseudomonas aeruginosa* PAO-MW1 was cultured in LB medium at 37 °C and maintained with 50 μg/mL tetracycline and 15 μg/mL mercury chloride (the latter strain was generously provided by Peter Greenberg from the Microbiology Core of the University of Washington Cystic Fibrosis Research). Stock solutions of various AHLs (C4-AHL, C6-AHL, C8-AHL, and 3-OXO-C12-AHL (C12-AHL)) were prepared in HPLC-grade CH3CN (Sigma-Aldrich, Gillingham, UK) and diluted in PBS to achieve a working concentration of 100 μM. These stock solutions were either used immediately or stored at −20 °C.

### 2.2. Fermentation and Extraction from Halorubrum saccharovorum CSM52

*H. saccharovorum* CSM52 was obtained from Kilroot Salt Mine, located in Carrickfergus, County Antrim, Northern Ireland. The strain was isolated from a saturated brine-pool and has been characterized and identified as a haloarchaea species using whole genome sequencing. *H. saccharovorum* CSM52 was cultured in Modified Payne Media plus 20% NaCl (MP20) (NaCl 250 g/L, MgSO_4_.7H_2_O 20 g/L, FeSO_4_.4H_2_O 0.036 g/L, MnCl_2_.4H_2_O 3.6 × 10^−3^ g/L, yeast extract 0.4 g/L, casein hydrolysate 0.2 g/L, trisodium citrate 0.3 g/L) for four weeks at 37 °C and 150 rpm.

The cultures (1000 mL) were centrifuged (10,000 rpm, 10 min), and the supernatant was extracted with 1000 mL dichloromethane (DCM, HPLC grade, Sigma, UK) following established methods for AHLs [18]. The crude organic layer was collected, dried over anhydrous sodium sulfate, and filtered. The aqueous phase was re-extracted twice more with DCM, and the organic layers were combined, dried, and transferred to a clean round-bottom flask. The volume was reduced using a rotary evaporator (Rotavapor R-114, Büchi, Newmarket, UK), and the crude extract was resolubilized in minimal CH_3_CN and dried under N_2_ gas. A control solvent extract was prepared from an uninoculated MP20 using the same procedure.

### 2.3. Thin Layer Chromatography (TLC) Overlay Assays

TLC overlay assays were conducted to characterize the crude extract based on its retention time relative to other known AHL molecules using a previously described assay [19]. Briefly, 1 μL of standard AHL solutions (100 μM) and 1 μL of the crude archaeal extract (50 mg/mL) were spotted onto RP-C18 TLC plates (Merck Ltd., Darmstadt, Germany). The plates were developed using a 60% MeOH: 40% H_2_O mobile phase, air-dried in a class-2 laminar flow microbiology cabinet for 10 min, and placed in sterile Petri dishes.

To perform the bacterial overlays, 1 mL of an overnight culture of *A. tumefaciens* was seeded in 10 mL of 0.5% molten LBA with X-gal (60 μg/mL). The plates were left at room temperature to solidify before incubation at 28 °C for 24–48 h. *A. tumefaciens* carries a plasmid that enables the AHL-inducible production of β-galactosidase (LacZ) activity, which converts X-gal into a green/blue pigment. *A. tumefaciens* is a versatile bioreporter strain that is capable of responding to a broad range of AHL molecules including 3-oxo-AHLs, C6-AHL, C8-AHL, C10-AHL, C12-AHL, C14-AHL, C6-3-hydroxy-AHL, C8-3-hydroxy-AHL, and C10-3-hydroxy-AHL [20].

For CV026, a similar method was used, adding 50 μL of the overnight culture to 10 mL of 0.5% molten LBA. C6-AHL is used as a positive control, and a media control served as a negative control. CV026 is a mutant strain of *C. subtsugae* that cannot produce C6-AHL but can respond to exogenous C6-AHL, C6-3-oxo-AHL, C8-AHL, C8-3-oxo-AHL, and C4-AHL, leading to violacein production.

### 2.4. Bioluminescence Assay

The bioluminescent reporter strains *E. coli* JM109 pSB536, *E. coli* JM109 pSB401, and *E. coli* JM109 pSB1142, which are designed to detect short-chain (C4 AHLs), medium-chain (C6-C8 AHLs), and long-chain AHLs (C10-C12 AHLs), respectively, were used to quantify bioactivity. These strains were inoculated directly from frozen stocks into 100 mL of LBB, with the appropriate antibiotic, and incubated overnight before being used in the experiments. Briefly, 20 μL of the crude extract (resolubilized in H_2_O and filtered through a 0.45 μM membrane filter) was added to 180 μL of a 1 in 100 dilution of an overnight culture of *E. coli*, resulting in a final extract concentration of 2.5 mg/mL. A media extract at the same concentration served as a media extract/untreated (UT) control. As positive controls, 20 μL of AHL standards (PBS, 100 μM) was added to 180 μL of a 1 in 100 dilution of an overnight culture of *E. coli*. The luminescence of the bioreporter strains was measured over 14 h at 37 °C for pSB401/pSB1142 and at 30 °C for pSB536 using a plate reader, and the luminescence values were normalized against the Optical Density (OD) at 550 nm (BMG Labtech Ltd., Aylesbury, UK).

### 2.5. Lactonolysis of AHLs

Briefly, 100 μL of crude extract (CH_3_CN, 50 mg/mL) was dried under an N_2_ stream and was resolubilized in 50 μL NaOH solution (5 mM) (equivalent to pH 11.6). as per Yates et al. [21]. To serve as a positive control, 20 μL of AHL standards (CH_3_CN, 100 μM) were resolubilized in 50 μL of 5 mM NaOH. The inactivation of the AHLs was confirmed by measuring the ability of the crude extract and AHL standards to induce X-gal conversion by *A. tumefaciens*.

### 2.6. Colorimetric Quantification of Bioactive Compound(s)

The method for the detection of AHL was developed by Yang et al. [22]. Briefly, 200 μL of *H. saccharovorum* CSM52 extract (resolubilized in H_2_O) was successively mixed with 250 μL of a 1:1 mixture of 2 M hydroxylamine (NH_2_OH) and 3.5 M NaOH (1:1% *v*/*v*). After mixing for a few minutes, 250 μL of a 1:1 mixture of 10% ferric chloride prepared in 4 M HCl: 95% EtOH was added. The reaction between the ester group of AHL and hydroxylamine (NH_2_OH) leads to the formation of hydroxamic acid in a basic solution. This hydroxamic acid forms a highly colored complex with ferric ions, which is quantifiable at 520 nm.

### 2.7. Pyoverdine and Pyocyanin Production

To assess cross-kingdom function, the extract was tested for its ability to induce Pseudomonal QS-dependent virulence factors (pyocyanin and pyoverdine) in PAO-MW1 and PAO1, following previously established methods [19]. Pyocyanin production is regulated by C4-AHL and further induced by C12-AHL, while pyoverdine is regulated by C12-AHL. The extract (25 mg/mL, resolubilized in H_2_O and filtered through a 0.45 μM membrane filter) was added to a 1:100 dilution of an overnight culture of PAO-MW1 in LBB, resulting in a final concentration of 2.5–0.625 mg/mL. Both C4-AHL and C-12AHL (PBS, 100 μM) were added at a 1:1 ratio for a final concentration of 10 μM, serving as a positive control, while PBS was used as a negative control. After incubation, culture supernatants were collected after centrifugation at 10,000 rpm for 10 min. Pyoverdine production was measured from the culture supernatant using a plate reader at ex. 400 nm and em. 460 nm. Pyocyanin absorbance in the supernatant was measured at 550 nm using a plate reader (BMG Labtech Ltd., Aylesbury, UK). Pyocyanin and pyoverdine values were normalized according to growth at OD_550_ after 24 h of growth.

### 2.8. Influence of the Extract on H. saccharovorum CSM52 Biofilm Biomass Development

The crystal violet assay is a well-established method for determining biomass/biofilm density in bacterial biofilms [23], and it has also been previously used for the assessment of archaeal biofilm biomass [24]. *H. saccharovorum* CSM52 biofilms were grown for 14 days using an MBEC (Innvotech^®^, Edmonton, AB, Canada) at 37 °C at 150 rpm. A total of 150 μL of 0.1 OD_550_ inoculum was added to each well. Crude extract of CSM52 (0.5 and 0.25 mg/mL in H_2_O), a QS inhibitor, (*Z*-)-4-bromo-5-(bromomethylene)-2(5H)-furanone (5-BF), and the extract (0.25 mg/mL), with 5-BF (50 μM), were incubated to *H. saccharovorum* CSM52 wells to determine their effect on biomass production. A media extract at the same concentration served as a media extract/untreated (UT) control. Following incubation, the lid was removed, and the liquid in each well was expelled using a wide bore pipette to prevent the disruption of the biofilm biomass. Each well was rinsed twice with 180 μL 30% NaCl solution to remove loosely attached cells. Then, 180 μL of 0.1% crystal violet (in 30% NaCl solution) was added to each well and stained for 1–2 h at room temperature. The wells were rinsed with 30% NaCl solution and then air-dried. The dye bound to the cells was resolubilized with 180 μL of 33% glacial acetic acid for 1–2 h at room temperature, and the concentration of crystal violet was determined at 590 nm using a plate reader (BMG Labtech Ltd., Aylesbury, UK).

### 2.9. Statistical Analysis

Statistical analysis was performed using one-way ANOVA and Dunnett’s post-test analysis or Tukey’s multiple comparisons tests. *p* values of * (*p* ≤ 0.05), ** (*p* ≤ 0.01), *** (*p* ≤ 0.001), and **** (*p* ≤ 0.0001) indicated statistical significance. GraphPad Prism 9.4.1 was used for all analyses. Mean values are based on replicates of at least three ± standard error.

## 3. Results

### 3.1. Induction of β-Galactosidase, Violacein Production, and Biolumienescence by the Crude Extract

Our results demonstrate that the crude extract induced β-galactosidase activity in *A. tumefaciens*, as evidenced by the production of green/blue pigment in soft agar overlays (Figure 1a). The Rf value of the extract (0.56) was comparable to that of the C4-AHL control (Rf = 0.56) and close to that of C6-AHL (Rf = 0.37), suggesting that the bioactive compound may be similar to C4-AHL or C6-AHL molecules or possess similar polarity.

To further investigate the structural similarity of the bioactive compound to AHL molecules, we assessed its ability to induce violacein production in *C. subtsugae* CV026, which is specifically responsive to exogenous C6-AHL and its homologs. Our results indicate that the extract did not induce violacein production (Figure 1b), suggesting that the bioactive compound is not structurally similar to C6-AHL.

We used the bioluminescent reporter strains *E. coli* JM109 pSB536 (responsive to C4 AHLs), *E. coli* JM109 pSB401 (responsive to C6–C8 AHLs), and *E. coli* JM109 pSB1142 (responsive to C10–C12 AHLs) to further characterize and quantify the bioactive compound present in the CSM52 extract. As shown in Figure 2, the CSM52 extract was able to induce bioluminescence in *E. coli* JM109 pSB536 but not in the other bioluminescent reporter strains. These results suggest that the crude extract possesses a bioactivity similar to that of C4-AHLs.

### 3.2. Investigating the Nature of the Bioactive Compound

AHL signaling molecules can be inactivated through lactonolysis by raising the pH to 11.6 or higher. We used this method for the preliminary structural elucidation of active compounds in the crude extract. Figure 3a shows that the crude extract retained activity towards *A. tumefaciens* when resolubilized at pH 11.6, suggesting the bioactive compound might be a modified AHL, as reported in methanogens [4], but it is unknown if these modifications can prevent lactonolysis. Alternatively, the bioa ctive compound could be a diketopiperazine (DKP), like the one identified by Paggi et al. from another haloarchaea capable of inducing *β*-galactosidase activity in *A. tumefaciens* [25].

We conducted colorimetric quantification for AHLs based on Yang et al. [22], which involved the formation of hydroxamic acids from esters by reaction with hydroxylamine in an alkaline solution. Figure 3b reveals no detectable amount of AHLs in the crude extract, and this assay, however, has limitations in terms of sensitivity for chain length and did not produce significant absorbance for 3-oxo-C12-AHL. These findings further support the hypothesis that the bioactive compound is either a modified AHL or a DKP.

### 3.3. Elucidating the Biological Function of the Extract from H. saccharovorum

We tested the bioactive compound for its ability to induce the AHL-dependent virulence factors pyocyanin and pyoverdine in *P. aeruginosa* PAO-MW1 (an rhlI and lasI double knockout) and wild-type PAO1. Figure 4a shows that the extract did not stimulate pyocyanin production, but it did induce dose-dependent pyoverdine production (Figure 4b). To determine if the extract was homologous to C4-AHLs, exogenous C12-AHL was co-incubated with the crude extract, and pyocyanin and pyoverdine production was measured. Figure 4c,d indicate that the extract did not induce pyocyanin production with C12-AHL, and pyoverdine production was still observed, suggesting no antagonism between the bioactive compound and C12-AHL.

We then tested the extract’s effect on pyocyanin and pyoverdine production in PAO1. Figure 4e,f reveal that the extract reduced pyocyanin production (*p* ≤ 0.01), while it increased pyoverdine production (*p* ≤ 0.01). The same bioactive compound could induce pyoverdine production and inhibit pyocyanin production, as previously demonstrated with 3-oxo-C12-AHL antagonizing C4-AHL [26]. These results indicate no homology between the bioactive compound from CSM52 and C4-AHL despite similar Rf values and show that the bioactive compound can induce QS-regulated pyoverdine production and inhibit pyocyanin production in both PAO-MW1 and PAO1. These findings appear to contradict the bioluminescence results suggesting C4-like activity and further imply that the bioactive compound is either a modified molecule or a DKP capable of interacting with QS pathways within *P. aeruginosa*.

Paggi et al. had previously proposed that extracellular protease production in *Natronococcus occultus* was QS-dependent [25]. We assessed the supernatant from CSM52 for proteolytic activity using azocasein over six weeks at seven-day intervals. However, our results (Appendix A) indicated that this observation could not be replicated in *H. saccharovorum* CSM52, as neither incubation time nor temperature (37 and 45 °C) promoted protease activity.

Previous studies have reported that antimicrobial production is QS-regulated in specific bacteria [27]. To assess the presence of antimicrobial production and QS regulation, we grew *H. saccharovorum* CSM52 for six weeks at 37 and 45 °C, with samples taken at 7-day intervals, and screened for antimicrobial production against *Haloferax volcanii* DS2. The results (Appendix A) showed no antimicrobial activity against the test strain, suggesting that other antimicrobials might be produced but are not displayed in this experiment.

To further investigate the bioactive compound’s potential role in biofilm formation, we incubated the extract with *H. saccharovorum* CSM52 for 14 days in an MBEC device, as the biofilm biomass was previously observed at this time point. Figure 5 shows a significant increase in biofilm biomass when CSM52 is co-incubated with 0.5 mg/mL (*p* ≤ 0.0001) and 0.25 mg/mL (*p* ≤ 0.0001) of the extract, while the QS inhibitor 5-BF did not affect biofilm biomass production. Interestingly, co-incubation with the 5-BF inhibitor and 0.25 mg/mL of the extract resulted in a relative decrease in biofilm biomass (*p* ≤ 0.0001), suggesting a potential interaction between the bioactive compound and the 5-BF inhibitor.

## 4. Discussion

This study aimed to explore QS-controlled production in *A. tumefaciens* and its potential role in cross-domain signaling between haloarchaea and bacterial pathogens. Our findings contribute to the growing body of research on QS signaling in archaea and its implications for interdomain interactions. We employed bioassays, bioreporter strains, and chemical characterization techniques to examine the bioactive compound in the extract. The main results can be summarized as: (1) *H. saccharovorum* CSM52 crude extract induces β-galactosidase activity in *A. tumefaciens*, suggesting the presence of QS signaling molecules; (2) the bioactive compound may be distinct from AHLs, possibly a chemically modified AHL or a DKP; (3) the extract influences *P. aeruginosa* virulence factor production, reducing pyocyanin levels and stimulating pyoverdine production; and (4) the extract increases biofilm biomass in *H. saccharovorum* CSM52, indicating a potential QS role in *Halorubrum* biofilm formation.

The initial characterization of the extract using TLC overlay revealed comparable polarity to C4-AHL and possibly C6-AHL. Previous studies with β-galactosidase expression in *A. tumefaciens* can be triggered by AHLs and DKPs [28], including the DKP cyclo-(L-prolyl-L-valine), from *Haloterrigena hispanica* [29]. Further characterization using *C. violaceum* CV026 suggested that the bioactive compound was structurally distinct despite a similar Rf value to C6-AHL. The extract activated pSB536, indicating some structural similarity to C4-AHL. DKPs from *P. aeruginosa* [28] have previously been observed to induce luminescence and compete for the same luxR binding site [30]. Cyclo-(L-prolyl-L-valine) also activated the pSB401, suggesting that these archaeal compounds may activate these bioreporters [29]. Therefore, the extract’s bioactivity may belong to either family of signaling molecules.

Further chemical characterization failed to detect quantifiable lactone ring/ester bonds or AHL-like compounds, suggesting that the bioactive compound could be a chemically modified AHL or a different signaling molecule, such as a DKP. Previous research has shown that the carboxylation of AHLs from the archaea *Methanothrix* (*Methanosaeta*) *harundinacea* is possible, but the effect on stability remains unclear [4]. The haloalkaliphilic archaea *N. occultus* and *Natrialba magadii* were reported to survive at high pH levels > 9 [31], suggesting they may produce compounds with modifications to improve stability in alkaline conditions with a pH level of >8.2 [21,32].

The extract potential similarity to C4-AHL was evaluated by assessing its influence on the production of pyocyanin and pyoverdine, virulence factors in *P. aeruginosa* controlled by the las and rhl QS systems. The lasI product directs the formation of C12-AHL [33], which, along with lasR, drives pyoverdine production [9], and other virulence genes including lasB, lasA, apr, toxA, and lasI itself [34,35,36,37]. The rhlI product catalyzes C4-AHL synthesis [38,39], which, together with rhlR, activates several secondaries such as pyocyanin, cyanide, and chitinase [40,41,42]. The las system controls the expression of rhlR and forms a hierarchy in QS [43,44]. The extract did not stimulate pyocyanin production when co-incubated with C12-AHL, suggesting a dissimilarly to C4-AHL. However, it reduced pyocyanin production in PAO1, indicating possible antagonistic interactions with native AI compounds. Antagonism between signaling molecules, such as 3-oxo-C12-AHL counteracting C4-AHL, has been reported [26]. Antagonism between signaling molecules, such as 3-oxo-C12-AHL counteracting C4-AHL, was observed here. The extract induced pyoverdine production in both PAO-MW1 and PAO1, with a more significant effect in PAO-MW1 due to the absence of endogenous AHLs. Co-incubation of the extract with C12-AHL further increased pyoverdine production, indicating a similar response to C12-AHL without antagonism. The impact on *P. aeruginosa* virulence factor production raises questions about the potential implications of interdomain signaling between haloarchaea and bacterial pathogens in complex microbiomes. Considering potential ecological implications, it is worth noting that both *Pseudomonas* and haloarchaea have been identified from hypersaline brine-pools [45]. This cohabitation raises intriguing questions about the possible roles of AHL-like signaling in these environments. While our study was conducted in laboratory conditions, it provides a springboard for future research to explore potential interdomain signaling in these natural habitats. Intriguingly, cross-domain interactions might not be limited to environmental contexts. A recent study found elevated levels of 2-hydroxypyridine, associated with *Methanobrevibacter smithii*, in Parkinson’s disease patients [46]. This emphasizes the need for further studies to elucidate the complex interactions between archaea and other microbes in various environments, including the human body.

To elucidate the function of bioactivity in haloarchaea, we investigated several phenotypes previously reported as QS-regulated. Protease production in *N. occultus* has been shown to be QS-dependent [25], and certain antimicrobials, such as bacteriocin from *Streptococcus pneumonia*, are QS-regulated [47]. However, *H. saccharovorum* CSM52 showed no significant proteolytic activity or antimicrobial activity against *Hfx. volcanii* DS2. While limited studies on archaea biofilms exist [48], the presence of persister cells [49] suggests that haloarchaea can adapt to environmental changes within biofilms, potentially through QS coordination. The incubation of the extract with *H. saccharovorum* CSM52 resulted in biofilm biomass increases at 0.5 mg/mL and 0.25 mg/mL concentrations. The interaction between the 5-BF inhibitor further suggests that the bioactive was potentially a modified AHL molecule [50].

Our findings build on previous reports of AHL-like molecules in haloarchaea [23], providing additional evidence for QS signaling in these organisms. The presence of a potential AHL-like compound in *Halorubrum lacusprofundi* detected by an *E. coli* GFP bioreporter assay [24] confirms a potential role in biofilms. These results support the notion that AHL-like molecules in *H. saccharovorum* CSM52 and *H. lacusprofundi* are related to biofilm upregulation and formation within the *Halorubrum* genus.

Although we have gained insights into the extract’s potential similarity to AHLs and its effects on *P. aeruginosa* virulence factors, the exact chemical identity remains unknown. Advanced analytical techniques, such as mass spectrometry and nuclear magnetic resonance, could be employed in future studies to elucidate its structure and mechanism of action.

## 5. Conclusions

Our study provides initial insights into QS signaling in haloarchaea and its potential role in cross-domain interactions with bacteria, highlighting the possible ecological and clinical implications of these findings. The bioactive compound, possibly a chemically modified AHL or DKP, can influence *P. aeruginosa* virulence factors and *H. saccharovorum* biofilm formation. These interactions could have significant impacts in cohabited environments such as hypersaline brine pools. Despite limitations such as the use of crude extracts, these results raise significant questions regarding the potential for archaea to enhance pathogen virulence and what role an anti-archaeal drug/disruption of the archaeome may have. Future research is necessary to explore archaeal QS signaling, molecular mechanisms underlying cross-domain interactions, and the ecological and clinical relevance of these observations. This will provide a more complete understanding of the potential roles of haloarchaea in health, disease, and their environments. Understanding these aspects will contribute to a more comprehensive view of haloarchaea’s role in health, disease, and environmental interactions.

## Figures and Tables

**Figure 1 microorganisms-11-01271-f001:**
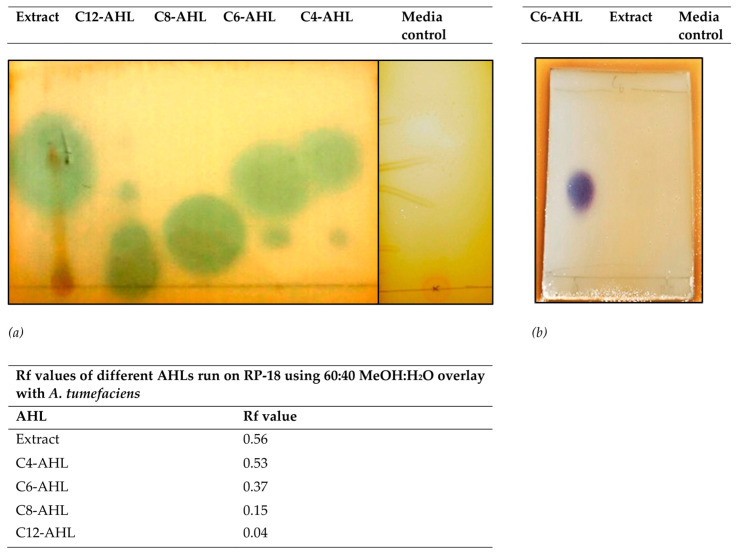
(**a**) *A. tumefaciens* TLC overlay assay for the detection of QS molecules from the crude extract. From the left lane: Crude extract under UV light (254 nm), followed by crude extract (50 mg/mL), C12-AHL (100 μM), C8-AHL (100 μM), C6-AHL (100 μM), C4-AHL (100 μM), and media control (50 mg/mL) following 48 h of incubation. The Rf value for the detected compounds is calculated below. (**b**) TLC overlay for AHL compounds using *C. subtsugae* CV026 with C6-AHL (left), crude extract (50 mg/mL) (middle), and media control (50 mg/mL) (right).

**Figure 2 microorganisms-11-01271-f002:**
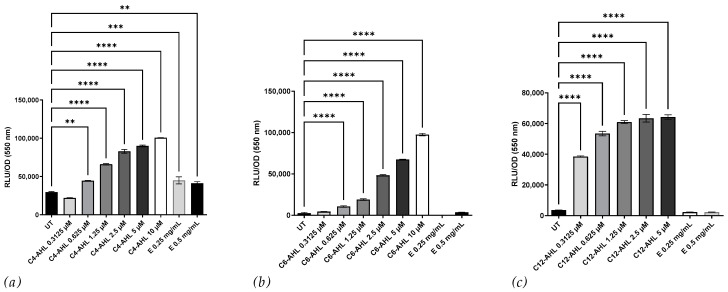
Effect of *H. saccharovorum* CSM52 crude, conditioned growth media extract on the bioluminescence reporter strains (**a**) *E. coli* JM109 pSB 536 (C4-AHLs), (**b**) *E. coli* JM109 pSB401 (C6-C8-AHLs), and (**c**) *E. coli* JM109 pSB 1142 (C10-C14-AHLs). The crude DCM extract (E) was resolubilized in H_2_O and filtered through a 0.45 μm filter, and the extract was tested at 0.25 and 0.5 mg/mL against each strain. C4-AHL, C6-AHLs, and C12-AHL were used as the positive control at a concentration from 0.3125 to 5 µM. A media extract served as the untreated (UT) control. Results are displayed as relative light units (RLU) normalized by the OD_550_. Asterisks indicate significant differences between the untreated and treated groups ** (*p* < 0.01), *** (*p* < 0.001), and **** (*p* < 0.0001) using one-way ANOVA and Dunnett’s post-test analysis *n* = 3).

**Figure 3 microorganisms-11-01271-f003:**
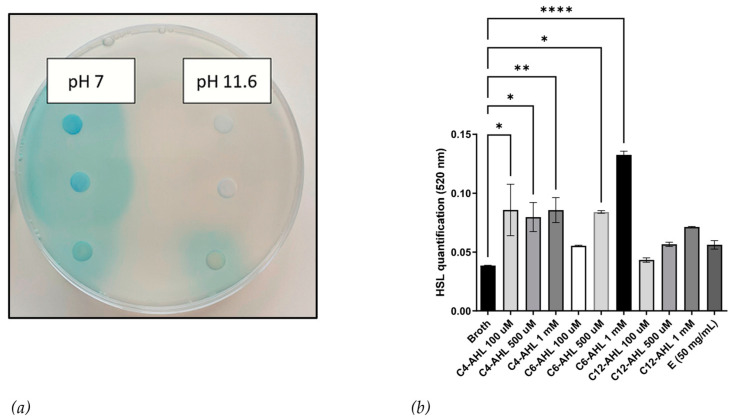
(**a**) CSM52 extract (E) was resolubilized in both pH 7 and pH 11.6; bioactivity was confirmed using *A. tumefaciens*. Left: (Top to bottom) C6-AHL, C12-AHL, and 1 mg/mL of extract resolubilized in pH 7. Right: (Top to bottom) C6-AHL, C12-AHL, and 1 mg/mL of extract resolubilized in pH 11.6. All AHLs were tested at a concentration of 10 μL/disc (100 μM). (**b**) Quantification of HSL determined by colorimetric analysis at 520 nm. Different concentrations of C4-AHL, C6-AHL, and C12-AHL were used to demonstrate the linear relationship between AHL concentration and quantification. From the supernatant of CSM52, no measurable amount of AHL compounds was detectable. Asterisks indicate significant differences between the untreated and treated groups * (*p* < 0.05),** (*p* < 0.01), and **** (*p* < 0.0001) using one-way ANOVA and Dunnett’s post-test analysis *n* = 3.

**Figure 4 microorganisms-11-01271-f004:**
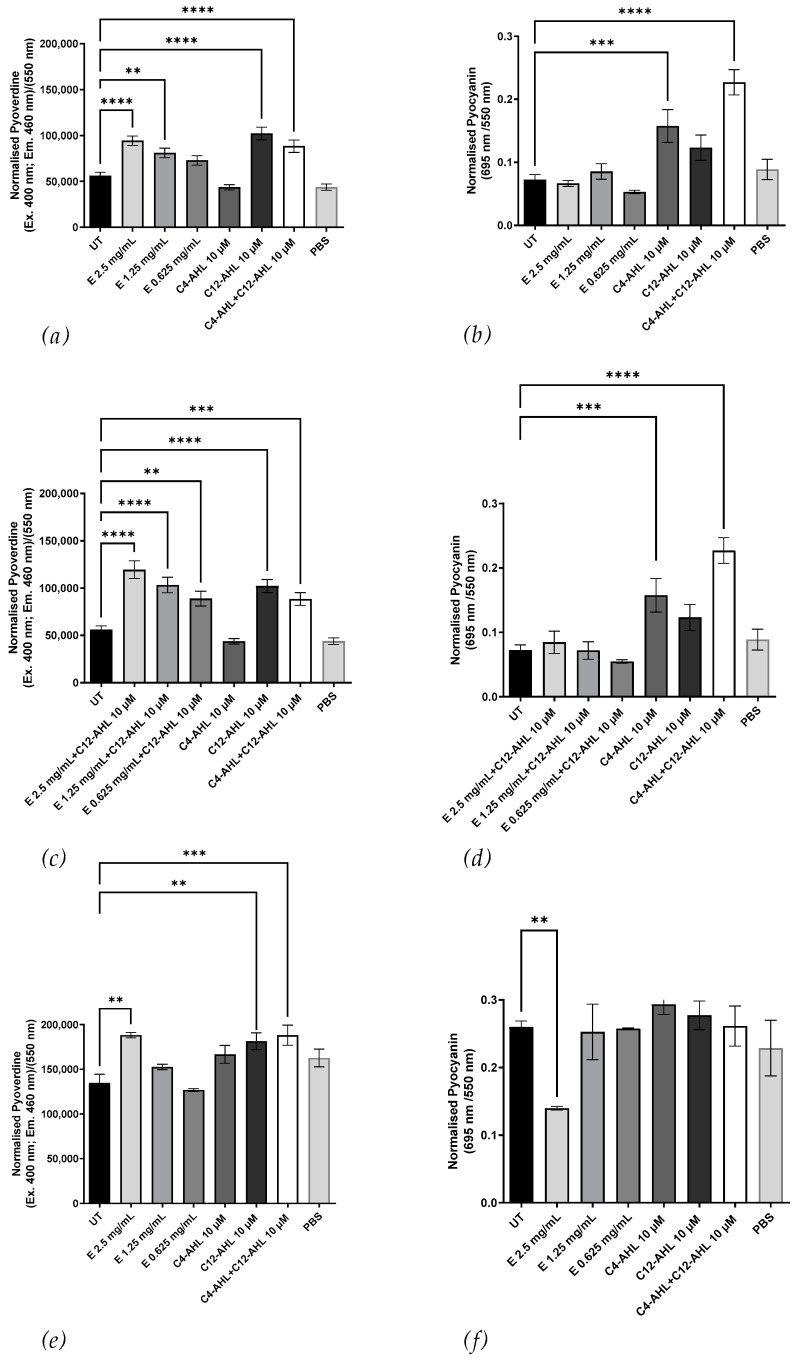
Effect of *H. saccharovorum* CSM52 extract (E) on (**a**) Pyoverdine production and (**b**) Pyocyanin production using the bioreporter PAO-MW1. Effect of the crude extract (E) with 10 μM C12-AHL on (**c**) Pyoverdine production and (**d**) Pyocyanin production using the bioreporter PAO-MW1. C4 and C12-AHL were used as positive controls (10 μM). Effect of the crude extract on (**e**) Pyoverdine production and (**f**) Pyocyanin production using PAO1. Results were normalized by OD_550_. Values are expressed as the mean and standard deviation of three replicates. Asterisks indicate significant differences between the untreated and treated groups ** (*p* < 0.01), *** (*p* < 0.001), and **** (*p* < 0.0001) using one-way ANOVA and Dunnett’s post-test analysis.

**Figure 5 microorganisms-11-01271-f005:**
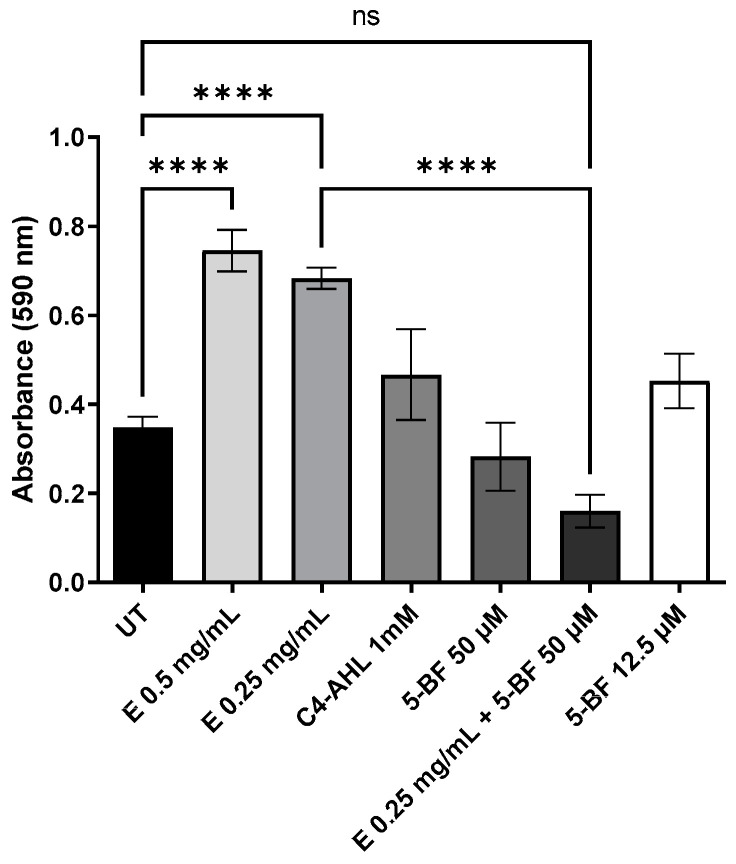
Biofilm biomass of *H. saccharovorum* CSM52 was measured by the absorbance of crystal violet staining and resolubilized in 33% acetic acid after the incubation of the culture for 14 days in an MBEC device at 37 °C. The presence of the extract (E) increased biomass production in *H. saccharovorum* CSM52. A media extract served as the untreated (UT) control. Values are expressed as the mean and standard deviation of three replicates. Asterisks indicate significant differences between the untreated and treated groups **** (*p* < 0.0001) using one-way ANOVA and Dunnett’s post-test analysis.

## Data Availability

Not applicable.

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
