# Peer review of "Quorum Sensing in Halorubrum saccharovorum Facilitates Cross-Domain Signaling between Archaea and Bacteria"

_microorganisms, 2023, doi:10.3390/microorganisms11051271_

Round 1
Reviewer 1 Report
Dear Editor,
I have reviewed the article "Production of cross-domain signalling molecules by halophilic archaea" and I believe that it has some significant shortcomings that prevent it from being suitable for publication in its current form.
Quorum Sensing in haloarchaea and archaea, in general, is a largely unexplored field, and it is therefore a topic of great scientific interest.
The authors investigated quorum sensing activity in the haloarchaeon Halorubrum saccharovorum CSM52 and detected signal molecules in the cell-free DCM extract using different bacterial bioreporters. This finding is certainly interesting, but it needs to be accompanied by chemical characterization of the detected signal molecules for publication in an international scientific journal. This is not the first report of quorum sensing activity in halophilic archaea, and lacking information on the chemical characterization of the molecules involved in QS activity means no advancement is made to scientific knowledge.
In addition, the study is presented as a cross-domain communication finding, which cannot be affirmed based on the reported data. Halorubrum saccharovorum CSM52 and Pseudomonas aeruginosa do not belong to the same microenvironment. Therefore, the activation of quorum sensing in this pathogen by the extracellular DCM extract of the H. saccharovorum CSM52 cannot have ecological significance. The induction of the production of two Pseudomonal QS-dependent virulence factors, pyocyanin and pyoverdine, by the H. saccharovorum CSM52 DCM extract gives further confirmation of the production of molecules involved in QS mechanisms but should be considered a preliminary result for promising biotechnological applications. This is not the first article to report an effect on the QS system of Pseudomonas aeruginosa of extracts/molecules from halophilic archaea, and therefore the novel factor is missing.
Regarding the inconsistencies to underline:
In the paragraph "Detection of bioactivity"
- Bioreporter CV026 has an optimal response with exogenous C6-AHL, but it also detects C6-3-oxo-AHL, C8-AHL, C8-3-oxo-AHL, C4-AHL.
- Regarding Agrobacterium tumefaciens NTL4(pZLR4), it is not indicated that it responds optimally to C8-3-oxo-AHL, but it also has positivity with 3-oxo-AHLs, C6-AHL, C8-AHL, C10-AHL, C12-AHL, C14-AHL, C6-3-hydroxy-AHL, C8-3-hydroxy-AHL, and C10-3-hydroxy-AHL.
If information on one bioreporter is given in a section, it should be given on all of them.
Paragraph "Influence of extract on H. saccharovorum CSM52 biofilm biomass development"
- It is unclear how, despite using an MBEC (Innvotech®, Calgary device), which as the advantage that biofilms are established on the pegs, the biofilm formed in the wells is measured.
Figure 1(a)
- For the TLC-overlay it is not easy to focus and take optimal photos, but the one in Fig.1(a) is of really low quality, and does not even frame the entire TLC.
A different elution solvent is indicated compared to the 'Materials and methods' section: MeOH:H2O 60:40 compared to MeOH:H2O 55:45.
Figure 1(b)
I do not understand the need to cover the discs on which the samples were loaded with large text boxes.
Additionally, the article suffers from some structural problems, such as the missing references (ref. 1 in the text) and duplicated references (10-36, 31-54), which suggests that the article was compiled without proper attention to detail. It's important to ensure that the article is thoroughly reviewed and carefully edited before submission to a scientific journal.
In conclusion, the study presents an interesting finding on quorum sensing activity in Halorubrum saccharovorum CSM52, but it falls short of being publishable without the proper chemical characterization of the detected signal molecules. Furthermore, the claim of cross-domain communication cannot be affirmed based on the reported data. The study should also address the inconsistencies in the text and the structural problems, as well as reference important previous work on "quorum sensing in extremophiles". I would recommend rejecting the article and giving the authors the opportunity to address these gaps and resubmit it in a more complete and appropriate form.
Sincerely
Author Response
Regarding the inconsistencies to underline:
In the paragraph "Detection of bioactivity"
- Bioreporter CV026 has an optimal response with exogenous C6-AHL, but it also detects C6-3-oxo-AHL, C8-AHL, C8-3-oxo-AHL, C4-AHL.
- Regarding Agrobacterium tumefaciens NTL4(pZLR4), it is not indicated that it responds optimally to C8-3-oxo-AHL, but it also has positivity with 3-oxo-AHLs, C6-AHL, C8-AHL, C10-AHL, C12-AHL, C14-AHL, C6-3-hydroxy-AHL, C8-3-hydroxy-AHL, and C10-3-hydroxy-AHL.
If information on one bioreporter is given in a section, it should be given on all of them.
Many thanks for this commentary, we have added these details on each strain and have changed this section and incorporated it into the TLC overlay section, and now reads as follows:
2.3. Thin Layer Chromatography (TLC) overlay assays
TLC overlay assays were conducted to characterize the crude extract based on its retention time relative to other known AHL molecules using a previously described assay [19]. Briefly, 1 μL of standard AHL solutions (100 μM) and 1 μL of the crude archaeal extract (50 mg/mL) were spotted onto RP-C18 TLC plates (Merck Ltd, Germany). The plates were developed using a 60% MeOH: 40% H2O mobile phase, air-dried in a class-2 laminar flow microbiology cabinet for 10 min, and placed in sterile Petri dishes.
To perform the bacterial overlays, 1 mL of an overnight culture of A. tumefaciens was seeded in 10 mL of 0.5% molten LBA with X-gal (60 μg/mL). The plates were left at room temperature to solidify before incubation at 28 °C for 24 - 48 H. A. tumefaciens carries a plasmid that enables AHL-inducible production of β-galactosidase (LacZ) activity, which converts X-gal into a green/blue pigment. A. tumefaciens is a versatile bioreporter strain that is capable of responding to a broad range of AHL molecules including 3-oxo-AHLs, C6-AHL, C8-AHL, C10-AHL, C12-AHL, C14-AHL, C6-3-hydroxy-AHL, C8-3-hydroxy-AHL, and C10-3-hydroxy-AHL [20].
For CV026, a similar method was used, adding 50 μL of the overnight culture to 10 mL of 0.5 % molten LBA. C6-AHL used as a positive control, and a media control served as a negative control. CV026 is a mutant strain of C. violaceum that cannot produce C6-AHL but can respond to exogenous C6-AHL, C6-3-oxo-AHL, C8-AHL, C8-3-oxo-AHL, and C4-AHL, leading to violacein production.
Paragraph "Influence of extract on H. saccharovorum CSM52 biofilm biomass development"
- It is unclear how, despite using an MBEC (Innvotech®, Calgary device), which as the advantage that biofilms are established on the pegs, the biofilm formed in the wells is measured.
Many thanks for this, we were erroneous in our description of the technique and have since amended it so it now states correctly the following:
‘Following incubation, the lid was removed, and liquid in each well was expelled using a wide bore pipette to prevent disruption of the biofilm biomass. Each well was rinsed twice with 180 μL 30% NaCl solution to remove loosely attached cells. Then, 180 μL of 0.1% crystal violet (in 30% NaCl solution) was added to each well and stained for 1 - 2 H at room temperature. The wells were rinsed with 30% NaCl solution and then air dried. The dye bound to the cells was resolubilised with 180 μL of 33% glacial acetic acid for 1 - 2 H at room temperature, and the concentration of crystal violet was determined at 590 nm using a plate reader (BMG Labtech Ltd., Aylesbury, UK).’
Figure 1(a)
- For the TLC-overlay it is not easy to focus and take optimal photos, but the one in Fig.1(a) is of really low quality, and does not even frame the entire TLC.
We have endeavoured to improve the quality of this figure in the resubmission.
A different elution solvent is indicated compared to the 'Materials and methods' section: MeOH:H2O 60:40 compared to MeOH:H2O 55:45.
Thank you for this comment, we have fixed this error so now both mobile phases state the correct composition of 60% MeOH: 40% H2O
Figure 1(b)
I do not understand the need to cover the discs on which the samples were loaded with large text boxes.
We agree with the reviewer and have added a TLC-overlay with this bioreporter instead to enhance the clarity of this figure.
Additionally, the article suffers from some structural problems, such as the missing references (ref. 1 in the text) and duplicated references (10-36, 31-54), which suggests that the article was compiled without proper attention to detail. It's important to ensure that the article is thoroughly reviewed and carefully edited before submission to a scientific journal.
We thank the reviewer for picking up on these duplicates and have since reviewed our references to remove these, and have added any missing references including the suggested reference ‘quorum sensing in extremophiles’. ‘However, the specific roles and mechanisms of QS in archaea, and our understanding of QS systems in lesser-studied groups, such as haloarchaea, remains limited [17] .’
We have redrafted the discussion and introduction to account for these structural problems.
Reviewer 2 Report
Comments to the authors for evaluating the following manuscript
Title: “Production of cross-domain signalling molecules by halophilic archaea”
Abstract
· In the abstract, I did not find the results or the material and methods of this work, it seem like a review article
Introduction
· The information and references in the introduction section are quite old, and very small there are new information about the QS in the last five years. Additionally, it failed to explore the main problems and the aim and hypothesis of this study. Therefore, I suggest the rephrasing of the introduction sections by adding more and more information
· This paragraph was repeated in several section ( abstract and introduction)
· “we investigated the crude supernatant extract from the haloarchaea, Halorubrum saccharovorum CSM52, for its ability to activate bacterial AHL-dependent QS phenotypes”
· The introduction section is ended by a brief of this work; meanwhile, it must end by the aim or hypothesis of this study
Material and methods
· The first paragraph contain many grammatical mistakes you have to rephrase this paragraph
· Please provide the source of H. saccharovorum CSM52 in this study
· Please add the abbreviation of Halorubrum saccharovorum in line 67
· Please add the abbreviation of the following microbes (Chromobacterium violaceum, Escherichia coli , Agrobacterium tumefaciens , Pseudomonas aeruginosa in the first part of material and methods then used the abbreviations
· Is the disc diffusion assay used to detect bioactivity? What is the meaning of detection of bioactivity? Do you mean antimicrobial activities?
· Results section
· The quality of all figures are very poor especially figure 1. Additionally the design of figure sections (plate) is very bad
· I feel that all the results is a description for each figure separately, it need layout
· It isnot suitable to added citation in your results such as “Yates et al. [21].” Line 235 to 282. It must added in discussion section
Discussion:
· It needs numerous modifications. It should focus on explaining and evaluating what you found (the main results), showing how it relates to the new researches
Conclusion section:
· It must be rephrased, conclusion section must provide us with the applied implication of your results in concise manner
· Please add at the end of the manuscript the limitations: what can’t the results and discussion tell us?
Author Response
Abstract
- In the abstract, I did not find the results or the material and methods of this work, it seem like a review article
We have updated the abstract and now it references several techniques and approaches used in the study. The full methods section in the paper has a more comprehensive account of the experimental procedures.
Introduction
- The information and references in the introduction section are quite old, and very small there are new information about the QS in the last five years. Additionally, it failed to explore the main problems and the aim and hypothesis of this study. Therefore, I suggest the rephrasing of the introduction sections by adding more and more information
- This paragraph was repeated in several section ( abstract and introduction)
- “we investigated the crude supernatant extract from the haloarchaea, Halorubrum saccharovorum CSM52, for its ability to activate bacterial AHL-dependent QS phenotypes”
- The introduction section is ended by a brief of this work; meanwhile, it must end by the aim or hypothesis of this study
Thank you for providing valuable feedback, we appreciate your suggestions and have made the following changes to address your concerns to the introduction, and have redrafted the entire introduction to avoid repetitions, and included aims/hypothesis to the study.
Material and methods
- The first paragraph contain many grammatical mistakes you have to rephrase this paragraph
We have addressed these mistakes and redrafted and thank the reviewer, it now reads:
The six QS-reporter strains used in this study were as follows: Chromobacterium violaceum CV026, which was cultured in LB medium at 28°C and maintained using 25 μg/mL kanamycin; Escherichia coli JM109 pSB536, cultured in LB medium at 30°C and maintained with 25 μg/mL ampicillin; E. coli JM109 pSB401, cultured in LB medium at 37°C and maintained with 20 μg/mL tetracycline; E. coli JM109 pSB1142, cultured in LB medium at 37°C and maintained with 20 μg/mL tetracycline (all strains were kindly provided by Paul William from the School of Life Sciences at the University of Nottingham). Additionally, Agrobacterium tumefaciens NTL4(pZLR4) ATCC BAA-2240 was cultured at 28°C in LB medium and maintained with 25 μg/mL gentamicin, while Pseudomonas aeruginosa PAO-MW1 was cultured in LB medium at 37°C and maintained with 50 μg/mL tetracycline and 15 μg/mL mercury chloride (the latter strain was generously provided by Peter Greenberg from the Microbiology Core of the University of Washington Cystic Fibrosis Research). Stock solutions of various AHLs (C4-AHL, C6-AHL, C8-AHL, and 3-OXO-C12-AHL (C12-AHL)) were prepared in HPLC-grade CH3CN (Sigma, UK) and diluted in PBS to achieve a working concentration of 100 μM. These stock solutions were either used immediately or stored at -20°C.
- Please provide the source of H. saccharovorum CSM52 in this study
Thank you for pointing out the missing information regarding the source of H. saccharovorum CSM52. I have now included this information in the manuscript as follows:
"H. saccharovorum CSM52 was obtained from Kilroot Salt Mine, located in Carrickfergus, County Antrim, Northern Ireland. The strain was isolated from a saturated brine-pool and has been characterised and identified as a haloarchaeal species using whole genome sequencing."
- Please add the abbreviation of Halorubrum saccharovorum in line 67
- Please add the abbreviation of the following microbes (Chromobacterium violaceum, Escherichia coli , Agrobacterium tumefaciens , Pseudomonas aeruginosa in the first part of material and methods then used the abbreviations
Many thanks for noting these errors, we have amended the manuscript with these changes. We hope these address your concerns.
- Is the disc diffusion assay used to detect bioactivity? What is the meaning of detection of bioactivity? Do you mean antimicrobial activities?
Thank you for your question regarding the use of the disc diffusion assay in our study.
Yes, in this study, the disc diffusion assay was employed to detect bioactivity, which refers to the ability of the crude supernatant extract from H. saccharovorum CSM52 to modulate bacterial quorum sensing (QS) phenotypes. By "detection of bioactivity," we mean evaluating the potential effects of the extract on QS-dependent processes, such as biofilm formation, or virulence factor production. Our primary aim was to assess the impact of the extract on QS mechanisms rather than antimicrobial activities. To clarify this point and avoid confusion, we have revised the respective section of the manuscript, and removed the disc diffusion data and corresponding methodology in favour of more TLC overlays.
- Results section
- The quality of all figures are very poor especially figure 1. Additionally the design of figure sections (plate) is very bad
We agree with the reviewer, and have uploaded an improved version of Fig 1 A and B.
- I feel that all the results is a description for each figure separately, it need layout
Many thanks for this suggestion. We have revised the results section of our manuscript to have a more organized and cohesive layout, rather than just describing each figure separately.
- It is not suitable to added citation in your results such as “Yates et al. [21].” Line 235 to 282. It must added in discussion section
Many thanks for this suggestion, we added this change to our discussion as per your recommendation.
Discussion:
- It needs numerous modifications. It should focus on explaining and evaluating what you found (the main results), showing how it relates to the new researches
We have made severe modifications to the discussion, it section now presents a balanced and nuanced analysis of the study's findings and implications. The additional references establish relevance within the field, elaborated limitations and considered biases introduce appropriate cautions, enhanced clarity improves coherence, and corrected grammar upholds professionalism. Please let me know if you have any final feedback or suggestions for improvement before incorporating this version into the final paper. I appreciate your guidance in strengthening this critical aspect of the work
Conclusion section:
- It must be rephrased, conclusion section must provide us with the applied implication of your results in concise manner
- Please add at the end of the manuscript the limitations: what can’t the results and discussion tell us?
We agree that the conclusion could be improved and have redrafted it to make it more concise, and added limitations.
‘Our study provides initial insights into QS signaling in haloarchaea and its potential role in crossdomain interactions with bacterial pathogens, highlighting the possible clin-ical implications of these findings. The bioactive compound, possibly a chemically modi-fied AHL or DKP can influence P. aeruginosa virulence factors and H. saccharovorum biofilm formation. Despite limitations such as the use of crude extracts, these results raise signifi-cant questions regarding the potential for archaea to enhance pathogen virulence, and what role an anti-archaeal drug/disruption of the archaeome may have. Future research is necessary to explore archaeal QS signaling, molecular mechanisms underlying cross-domain interactions, and the clinical relevance of these observations. This will pro-vide a more complete understanding of the potential roles of haloarchaea in health and disease. Understanding these aspects will contribute to a more comprehensive view of haloarchaea's role in health and disease.’
Reviewer 3 Report
The paper by Thomas P. Thompson et al is devoted to signalling molecules in halophilic archaea.
The idea is publication-worth, the paper is well written, experiments are well done and data are well discussed. I suggest to accept the paper with minor revision of the figures to make them more representative.
Please merge multiple citations in one blok, i.e. [1,2] instead [2] [3]
please refer in text to figure parts, like Fig 1A, Fig 1B etc
The quality of Fig 1 a is too low
Fig 2. Explain in caption that C4-ahl, c10.. and c12 were used as a positive control. The figure lacks 1 mg/mL of extract, while it is mentioned in caption. Explain that E means the extract (in all figures)
Fig 3 a is difficilut to see the effect since abbreviations overlap the plate. I suggest to remove the white background and border in text boxes
Author Response
Q: Please merge multiple citations in one blok, i.e. [1,2] instead [2] [3]
Many thanks for noting this, we have since updated the citations to reflect this suggestion.
Q: please refer in text to figure parts, like Fig 1A, Fig 1B etc
We have also changed these changes to the figure details.
Q: The quality of Fig 1 a is too low
We have endeavoured to improve the quality of this figure in the resubmission.
Q: Fig 2. Explain in caption that C4-ahl, c10.. and c12 were used as a positive control. The figure lacks 1 mg/mL of extract, while it is mentioned in caption. Explain that E means the extract (in all figures)
Thank you for this observation. This figure now reads:
Figure 2. Effect of H. saccharovorum CSM52 crude, conditioned growth media extract on bioluminescence reporter strains (a) E. coli JM109 pSB 536 (C4-AHLs), (b) E. coli JM109 pSB401 (C6-C8-AHLs) and (c) E. coli JM109 pSB 1142 (C10-C14-AHLs). The crude DCM extract (E) was resolubilised in H2O and filtered through a 0.45 μm filter, and the extract was tested at 0.25 and 0.5 mg/mL against each strain.C4-AHL, C6-AHLs, and C12-AHL were used as positive control at a concentration from 0.3125 to 5 µM. Results were displayed as relative light units (RLU) normalised by the OD550.
Q: Fig 3 a is difficilut to see the effect since abbreviations overlap the plate. I suggest to remove the white background and border in text boxes
We agree, with this and have removed the boxes from the figure to make Fig 3A more clear
Reviewer 4 Report
The work of Thompson T.P., Busetti A., Gilmore B.F. is devoted to the study of the signal molecule produced by halophilic archaea. This study is important because it demonstrates the relationship between the archaea and bacteria domains, and it also expands the knowledge of quorum sensing in archaea. The results of this study are not only valuable for basic science, but they can also be used in clinical practice. However, the work has some shortcomings.
1. In the Introduction, I would like to learn more about H. saccharovorum: what kind of object is it, where does it live, what is its peculiarity?
2. The purpose of the study is not spelled out in the work.
3. The strain Chromobacterium violaceum CV026 was reclassified and now it is called Chromobacterium substsugae CV026 (Harrison A.M., Soby S.D. Reclassification of Chromobacterium violaceum ATCC 31532 and its quorum biosensor mutant CV026 to Chromobacterium subtsugae // AMB Express. 2020. No. 1. P. 202.).
4. Line 103. MP20 – it is not clear what kind of object this is.
5. Section 2.2. “Fermentation and extraction from H. saccharovorum CSM52” is written in general terms without specifics: what volumes were worked with, how much was taken, transferred, etc.
6. Fig. 1(a) – I would like more clarity of the image.
7. Fig. 1(b) и 3(a) – are oversized captions covering important parts of the drawings.
8. Fig. 2; 3(b); 4; 5 – does not have a designation of what "E" is.
In general, despite the lack of unambiguous conclusions about the nature of the H. saccharovorum autoinducer, as well as minor shortcomings, the work is relevant and can be recommended for publication.
Author Response
- In the Introduction, I would like to learn more about H. saccharovorum: what kind of object is it, where does it live, what is its peculiarity?
Thank you for pointing out the missing information regarding the source of H. saccharovorum CSM52. I have now included this information in the manuscript as follows:
"H. saccharovorum CSM52 was obtained from Kilroot Salt Mine, located in Carrickfergus, County Antrim, Northern Ireland. The strain was isolated from a saturated brine-pool and has been characterised and identified as a haloarchaeal species using whole genome sequencing."
- The purpose of the study is not spelled out in the work.
We have updated the introduction, so it covers this:
‘To tackle this knowledge gap, we aim to investigate the potential presence and role of QS systems in haloarchaea, with a specific focus on Halorubrum saccharovorum CSM52. The hypothesis of this study is that H. saccharovorum CSM52 produces extracellular signaling molecules that can activate bacterial AHL-dependent QS phenotypes, indicating a possi-ble involvement in interspecies communication and coordination. Furthermore, we will attempt to link this extract activity with a phenotype associated with AI production and investigate the potential ecological implications and functional roles of these signaling molecules in the context of haloarchaeal biology. By expanding our understanding of QS systems in haloarchaea, we hope to uncover novel communication mechanisms and po-tential applications in biotechnology, medicine, and environmental sciences.’
- The strain Chromobacterium violaceum CV026 was reclassified and now it is called Chromobacterium substsugae CV026 (Harrison A.M., Soby S.D. Reclassification of Chromobacterium violaceum ATCC 31532 and its quorum biosensor mutant CV026 to Chromobacterium subtsugae // AMB Express. No. 1. P. 202.).
Thank for pointing this out, we have now changed this, so this strain is referred to as C. subtsugae CV026 throughout the manuscript instead.
- Line 103. MP20 – it is not clear what kind of object this is.
This is clarified with this: The strain was isolated from a saturated brine-pool and has been characterised and identified as a haloarchaea species using whole genome sequencing. H. saccharovorum CSM52 was cultured in Modified Payne Media plus 20% NaCl (MP20) (NaCl 250 g/L, MgSO4.7H2O 20 g/L, FeSO4.4H2O 0.036 g/L, MnCl2.4H2O 3.6×10-3 g/L, yeast extract 0.4 g/L, casein hydrolysate 0.2 g/L, trisodium citrate 0.3 g/L) for four weeks at 37° C at 150 rpm.
- Section 2.2. “Fermentation and extraction from H. saccharovorum CSM52” is written in general terms without specifics: what volumes were worked with, how much was taken, transferred, etc.
We have admenedd this now so it states the following
‘Cultures (1000 mL) were centrifuged (10,000 rpm, 10 min) and the supernatant was extracted with 1000 mL dichloromethane (DCM, HPLC grade, Sigma, UK) following es-tablished methods for AHLs [18]. The crude organic layer was collected, dried over anhydrous sodium sulfate, and filtered. The aqueous phase was re-extracted twice more with DCM, and the organic layers were combined, dried, and transferred to a clean round-bottom flask. The volume was reduced using a rotary evaporator (Rotavapor R-114, Büchi, UK), and the crude extract was resolubilized in minimal CH3CN and dried under N2 gas. A control solvent extract was prepared from an uninoculated MP20 using the same procedure.’
- Fig. 1(a) – I would like more clarity of the image.
We have endeavoured to improve the quality of this figure in the resubmission.
- Fig. 1(b) и3(a) – are oversized captions covering important parts of the drawings.
We agree, with this and have removed the boxes from the figure to make Fig 3A more clear
- Fig. 2; 3(b); 4; 5 – does not have a designation of what "E" is.
We have added these extra details to the figure legends.
Round 2
Reviewer 2 Report
All figures in this manuscript are very poor and it isnot suitable to use marker pin to mark the figures which will be published in web of science journals
Author Response
Many thanks for these corrections to our article.
We have addressed all the recommended and thank the reviewers and editors for their invaluable experience and feedback. We believe it has significantly improved the quality of the manuscript.
